# Improvement of Soil Microbial Diversity through Sustainable Agricultural Practices and Its Evaluation by -Omics Approaches: A Perspective for the Environment, Food Quality and Human Safety

**DOI:** 10.3390/microorganisms9071400

**Published:** 2021-06-28

**Authors:** Marta Bertola, Andrea Ferrarini, Giovanna Visioli

**Affiliations:** 1Department of Chemistry, Life Sciences and Environmental Sustainability, Università di Parma, Parco Area delle Science 11/a, 43124 Parma, Italy; 2Department of Sustainable Crop Production, Università Cattolica del Sacro Cuore, Via E. Parmense 84, 29122 Piacenza, Italy; andrea.ferrarini@unicatt.it

**Keywords:** soil microbiome, functional microbial diversity, ecosystem services, sustainability, soil health, food safety

## Abstract

Soil is one of the key elements for supporting life on Earth. It delivers multiple ecosystem services, which are provided by soil processes and functions performed by soil biodiversity. In particular, soil microbiome is one of the fundamental components in the sustainment of plant biomass production and plant health. Both targeted and untargeted management of soil microbial communities appear to be promising in the sustainable improvement of food crop yield, its nutritional quality and safety. –Omics approaches, which allow the assessment of microbial phylogenetic diversity and functional information, have increasingly been used in recent years to study changes in soil microbial diversity caused by agronomic practices and environmental factors. The application of these high-throughput technologies to the study of soil microbial diversity, plant health and the quality of derived raw materials will help strengthen the link between soil well-being, food quality, food safety and human health.

## 1. Introduction

Soil is one of the key elements for supporting life on Earth. Indeed, almost half of the Sustainable Development Goals (SDGs) are linked to soil [1]; most can be pursued through sustainable land use and soil quality improvement, which can be defined as “the capacity of soil to function as a vital living system to sustain biological productivity, promote environmental quality and maintain plant and animal health”, where animal health includes human health too [2]. More precisely, soil quality should be expressed in terms of the soil’s capacity to deliver the multiple ecosystem services, which are provided by soil processes and functions [3]. The concept of soil quality has been integrated within the more holistic concept of soil health [4]. Soil health is an integrative property of agroecosystem management [5] and is commonly assessed through the combination of a set of physical, chemical and biological indicators representative of essential soil-based ecosystem services [3,6,7,8].

Nevertheless, in recent years several studies have shown that anthropogenic activities, such as agricultural intensification and land use change, as well as climate change, are reducing soil capacity to perform fundamental processes and functions [9,10]. This has led to increasing concern that reduced biodiversity in soils may impact negatively on many ecosystem functions and services [11,12]. 

Healthy soils are essential to support food production for the growing human population [13]. Soils characterized by active microbes and close interactions between fauna and plants bolster efficient nutrient cycling, pathogen resistance and overall crop health, and enhance long-term ecosystem stability given the future global change conditions [12]. In particular, soil microbes are the main organisms that sustain plant biomass production and plant health [14,15]. Several studies have shown that both direct and indirect manipulation of plant root and soil microbial communities appear to be a propitious strategy to enhance food crop yield, its nutritional quality and safety with beneficial effects for human and environmental health [16,17].

However, we ought to bear in mind the complexity of the relationships between biodiversity, services and functions that support ecosystems well, as these are still poorly understood. In general, there is evidence that soils with greater levels of biodiversity are more resistant to environmental disturbances and are consequently more resilient than the ones with reduced levels of biodiversity [18]. In addition, some processes that provide ecosystem functions are carried out by a greater number of species or groups of organisms (e.g. organic matter degradation), whereas other processes involve fewer species or groups of organisms (e.g. atmospheric nitrogen fixation) and so are more easily compromised by ecological disturbances [19]. Functional diversity may therefore be deemed a more appropriate way to consider the biotic status of soils than biodiversity *per se* [20].

In this regard, microbial communities offer a great potential to assess the functional biodiversity in soils because they are ubiquitous, are actively involved in biological functioning and ecosystem services provisioning, and are highly sensitive to environmental changes in terms of modifications in biomass, structure/diversity and activity [21,22,23]. The development of new molecular approaches called -Omics have recently allowed the characterization of the overall microbial genetic and functional diversity through the high throughput analysis of DNAs (genomics), RNAs (transcriptomics), proteins (proteomics), enzymes activities (enzymomics) or metabolites (metabolomics) [21]. 

This review paper aims to investigate the role and management of soil microbial diversity in sustaining the production of high-quality, safe food through -Omics approaches (Figure 1). It includes (i) discussion of the potential of high-throughput molecular techniques to evaluate genomic and functional soil microbial diversity (Section 2), (ii) analysis of the use of integrated management of soil microbial diversity to sustain crops productivity (Section 3), (iii) review of the current evidence on the links between soil microbial diversity, and food quality and safety (Section 4), and (iv) new insight into the relations between soil, plant, and human health (Section 5).

## 2. Using-Omics to Assess Soil Microbial Diversity

Recently, soil microbial abundance, diversity, activity and functional potential have been measured through high-throughput -Omics procedures, such as genomics, transcriptomics, proteomics, enzymomics and metabolomics [21,24] (Table 1). In particular, the advent of next generation sequencing techniques, such as complete genome shotgun sequencing, high-throughput sequencing and single molecule long-read sequencing, has allowed the identification of the microorganism communities present in soil, while the other –Omics approaches previously mentioned, have allowed the investigation of these communities’ biological functions [25,26,27]. There is an increasing interest toward –Omics approaches because, tighter with molecular techniques, they have overcome some of the limitations of cultivation-based studies [28]. Different authors have advocated for next-generation experiments to link the soil microbial community structure and the soil food web to different ecological processes, such as nitrogen fixation, litter decomposition and plant productivity [29,30,31]. By applying different -Omic approaches to the same target, microbial community diversity can be linked to ecological processes, ecosystem services and potentially food quality (Figure 1).

### 2.1. Soil Nucleic Acid High-Throughput Sequencing Technologies

Since the total microbial diversity of complex communities found in soil is hardly captured by culture techniques, molecular techniques capable of analyzing DNA directly are often employed. The most commonly utilized analysis is based on the amplification of bacterial 16S rRNA genes and fungal ITS region from total soil or rhizosphere extracted DNA, and the resulting PCR products are sequenced to identify the bacterial and fungal species present [32]. With the advent of high-throughput sequencing like pyrosequencing, Illumina/Solexa sequencing and Ion torrent sequencing platforms, millions of sequences can rapidly be obtained, thus providing a precise picture of a soil sample’s microbial diversity and also allowing the simultaneous comparison of microbial diversity in multiple soil samples [32].

The functional microbial diversity can be assessed by shotgun metagenomics sequencing, an emerging molecular method which enables to link the community structure with possible soil functional processes [33]. This methodology has improved understanding of the strategies adopted by microorganisms to thrive in different environmental conditions. Shotgun metagenome sequencing workflow started with random DNA fragmentation and adapter ligation like whole genome sequencing. The taxonomic analysis consists in filtering and evaluating a reference database containing whole genomes or specifically designed marker genes in order to create a taxonomy profile. 

A second functional microbial diversity high-throughput sequencing approach is based on the analysis of the transcriptome of a soil sample, which provides a direct vision of the metabolic activity through the gene expression profiles of its microbial community. The generation of raw transcriptome data involves purifying fine RNA from soil, transforming the RNA into complementary cDNA, fragmenting it to build a library and using a platform to sequence it [34].

### 2.2. Soil Metaproteomics 

Soil metaproteomics consists of the dissection of a complete protein profile derived from the microbial communities present in the soil [35]. It is also defined as a functional genomics approach since it helps to find the metabolic active microorganisms in soil. The activities of the soil microbial community depend on many external environmental stimuli. Thus, evaluating changes in protein expression patterns in soil could be a useful indicator for understanding the involvement of specific taxa in soil processes [35,36]. Metaproteome can be analyzed using different biochemical methods, depending on the type of information and level of resolution required. A non-targeted ‘protein map’ of a soil community can be obtained by separating the proteins through one- or two-dimensional gel electrophoresis (1DE) or (2DE). The proteins of interest can then be extracted from the gels, digested to peptides by proteolytic enzymes like trypsin which are later analyzed through mass spectrometry (MS) or tandem MS analysis [21]. The gel-free shotgun metaproteomic strategy is the most utilized method for soil proteomics. It is based on the digestion of the entire proteome into a complex peptide mixture by proteases. A combination of liquid chromatography (LC) and high-resolution mass spectrometry (HRMS) techniques [37] is then used to separate the complex peptide mix. A reliable metaproteomic investigation requires thorough computational elaboration in order to convert the raw spectral data into peptide sequence followed by protein inference. For this purpose, a database tailored for protein identification set up with information obtained by soil metagenome high throughput sequencing helps combine phylogenetic and functional microbial community data [38,39].

### 2.3. Soil Enzymomics

Enzymomics (-Omics of enzymes) is an emerging discipline in clinical studies [40] with still few applications in agri-environmental sciences. Due to the development of soil proteomics, it is increasing the interest to have extracted efficiently of the enzymes from soil matrix. Enzyme activity has long been used as a simple, effective and sensitive bioindicators to detect variations of soil biochemical activity at both laboratory and field level [41]. Soil hydrolytic enzymes, such as phosphatases, glucosidases, xylosidases, cellulases, sulfatases, chitinases and proteases, play a pivotal role in soil organic matter (SOM) decomposition, transformation, and mineralization [20]. SOM, if seen as a continuum of progressively microbially-processed organic compounds [42,43,44], it is at the core of the supply of regulating and supporting ecosystem services essential to the health of agroecosystems [45]. Variation in enzymatic activities is a useful first bioindicator of the change in the metabolism of the whole microbial community [46,47].

However, for a satisfactory evaluation of soil biochemical activity, a sufficient number of enzyme activities is usually necessary. Studies at ecosystem-level can involve the processing of many parameters on a large number (hundreds) of samples, which highlight the need for high-throughput procedures especially when assessing hydrolytic enzymes activities. The introduction of microplate-based assay [48] greatly improved the throughput and sensitivity, due to the use of fluorogenic substrates. This opened a new era for soil enzymes research by increasing the knowledge of enzymes production and stabilization mechanisms [49], the impact of land use and environmental controlling factors on enzyme activities [50], the origin of their spatial heterogeneity [51] and their usefulness within the broader context of soil microbial ecology [41]. However, little has been done to improve enzyme extraction in the view, from one side, of solving the current bottleneck of soil proteomics [52], and from the other side, of informing the structure–activity relationships into Earth system models [53]. As a result, the need emerged to improve enzyme extraction methods to perform high-throughput assay on clear extracts with no enzymes physically or chemically protected, or humus–enzyme complexes remaining active or most importantly with low interference of free enzymes (quenching) with humic substances [54]. Interestingly, [55] found that a protein (bovine serum albumin, BSA) coupled to a detergent (Triton X-100) greatly improved extraction of enzymes from soil. Not less important, co-extraction of humic substances was much lower compared to traditional procedures and extraction time was very low. Enzymes can be efficiently desorbed by heteromolecular exchange with a bead-beating procedure [54] that improves the extraction yields and greatly shortens the extraction time as shown also for dsDNA quantitation [56] as soil microbial biomass indicator [57]. Ferrarini et al. showed that the assay can be performed on 384-well microplates using fluorogenic substrates and a number of enzymes (>20) can be assayed by using the same soil extract [47]. This desorption-based method shown to be able to detect perturbations in different soils and climate [58] and among different land use and cropping systems [59,60] at statistically significant level when other slurry-based techniques did not. This detection and discrimination capability is not a small matter because, although subject of potential criticisms over the soil-specific extraction efficiency for some enzymes, it covers potentially an infinite spectrum of key soil enzymes (fluorogenic substrates are now commonly available for all enzymes classes) involved in the SOM decay continuum, from breakdown of complexes substrates until ultimate steps of mineralization of plant available nutrients. Soil enzymomics based on desorption methods ultimately offers the possibility to analyze via multivariate data analysis a highly representative enzymes dataset in order to provide “fit for purpose” advice to support the selection of the set of samples to be sequenced with other -Omics techniques. An example of the usefulness of such approach is given in [61] where samples for proteomics sequencing has been carried selected on the basis of enzyme patterns differentiation in rhizosphere soil where different biostimulants have been applied to maize seeds. Similar approaches are used to understand crop effects and help selecting representative samples to be sequenced for PGPR isolation in bioremediation studies of heavy metal contaminated soil and sediments [62].

### 2.4. Soil Metabolomics

The term ‘metabolome’ refers to the total metabolites in an organism or environment. Primary and secondary metabolites are produced by microorganisms present in an environment, such as soil following external stimuli. Metabolome-based approaches can be applied to dissect interactions between soil microbe and plant root structures because metabolites can determine microbial food webs, regulate soil chemistry, change microbial gene expression, and even act as info-chemicals to mediate microbe-to-microbe interactions [63,64]. Soil metabolomic analyses begin with sample preparation, detection and quantification through different chromatographic techniques (liquid chromatography and gas chromatography) and systems such as mass spectrometry and nuclear magnetic resonance [65]. Liquid chromatography MS is widely used among MS technologies [65]. With mass spectrometry, spectra are obtained consisting of a set of peaks that can be used to analyze and quantify metabolites. The spectra acquired from a given sample are analyzed through the use of databases, thus enabling automated analysis and the generation of metabolomic profiles [65]. Metabolomics along with bioinformatics tools and databases could enable a better understanding of microbial community, their catabolic pathways and genes responsible for encoding catabolic enzymes. Presently, few studies have been reported on the soil metabolome mainly addressing variations in the rhizosphere metabolites of different crops [66,67,68] and in investigating the global metabolic response of soil to nanomaterials and organic contaminants exposure [69].

## 3. Targeted and Untargeted Approaches to Soil Microbial Diversity Management

Agronomic management, which increase the soil microbial community, can be promising strategies to obtain agricultural systems that are more productive, resource-efficient, resilient and adaptive to global changes, minimizing environmental impacts [12]. Several studies have highlighted that it is possible and advantageous to address future needs by transitioning from conventional intensification of agriculture to a food production system based on “ecological intensification”; this means that soil microorganism enrichment can be effectively exploited as a nature-based solution to maintain high productivity levels by optimizing soil ecosystem services, while reducing the reliance on external input and minimizing adverse effects on the environment [70,71,72,73]. In this regard –Omics approaches can help to further understand the link between soil microbial diversity, its community composition and abundance, and ecological functions provided; hence, highlighting the benefit of ecological intensification (Table 1).

**Table 1 microorganisms-09-01400-t001:** -Omics approaches applied to soil microbial diversity studies.

Reported Organisms	Molecular Technique	Inference	Reference
Mycorrhizal fungi (AMF)	Metabolomics	Enhanced plant response to water stress, modulation of oxidative stress conditions and increased production of phytohormone	[74]
Endophytic fungi	Metabolomics	Increased plant and fruit yields thanks to improved production of phytohormones	[75]
Bacteria	16S rDNA gene pyrosequencing	Biofertilizers have effects on plant growth, rhizospheric microbes and native microbial communities	[76]
Bacteria and fungi	high-throughput pyrosequencing of bacterial 16S_V1-V3_ and fungal ITS_2_ of the ribosomal DNA operon	Organic farming increases microbial soil microbial diversity	[77]
Rhizospheric Bacteria	high-throughput sequencing of bacterial 16S rDNA gene amplicon	Pepper plants recruit beneficial microbes more efficiently under organic management, thereby increasing soil disease suppression	[78]
Bacteria	16S rDNA gene next generation sequences	Organic farming increases microbial soil microbial diversity	[79]
Bacteria	Metaproteomic associated with 16S rDNA genes sequencing	Biostimulant enhances the beneficial activity of microbes on plant growth	[61]
Bacteria and archaea	RT-PCR and pyrosequencing of 16S rDNA genes	No differences between conventional and organic farming on the composition of microbial communities	[80]
Rhizospheric bacteria	16S rDNA V_3_ region gene sequence	Biostimulant increases rare bacterial taxa, some of which involved in plant growth and pathogen resistance	[81]
Bacteria	16S rDNA gene sequences	Tritordeum cv. hire beneficial microbes more efficiently under organic management	[82]
Bacteria	Amplification of 16S rDNA V_3_-V_4_ regions and high-throughput sequencing	Rotary tillage and straw returning increase bacterial diversity	[83]

Two main approaches can be recognized in the management of the soil microbial diversity in agricultural systems (Figure 1):An untargeted approach based on agricultural practices (increased landscape diversity, complexity and connectivity between the natural ecosystem and agricultural fields; low-input management practices, such as organic farming; strategic crop rotation; intercropping; cover crops; agroforestry; minimum tillage with residue retention; green manures) applied to soil in order to increase biodiversity in favor of multifunctionality, resilience and adaptability to environmental changes [9]; it is a long-term strategy that takes into consideration the complexity, complementarity and self-regulation of soil biota.A targeted approach based on the knowledge of soil–plant interactions for specific crops which considers the biotechnological application of microorganisms such as biofertilizers, biostimulants, biopesticides or bioherbicides. This strategy still requires further investigation since the effects of applying specific microorganisms to a native microbial community is not fully understood yet.

### 3.1. Untargeted Approach

#### 3.1.1. Organic Farming

Organic farming is a low-input agricultural system that uses ecology-based pest control and biological fertilizers to sustain crop productivity. This practice and the derived products are regulated and certified by international and national institutional bodies along the whole supply chain [84,85]. Organic farming practices are known to have numerous ecological benefits. However, some studies have revealed that organic farming generally reduces crop yields compared to conventional farming [86,87,88]. This means that more cultivable land is needed to obtain the same amount of product as with conventional systems, which entails impacting on forests and other natural habitats. Nonetheless, organic farming appears to be more competitive under stress conditions and exhibits higher spatial and temporal stability [88,89]. For example, organically managed crops produce higher yields under drought stress, and up to 70–90% more under severe stress [87]. This derives from improved soil structure, higher SOM concentrations, better soil aggregation and water retention, greatest nutrients availability and higher soil food web biomass [87,88].

Data available on soil microbial diversity in organic and conventional farming do not appear to agree. Recent meta-analysis has shown that soil microorganisms usually react positively to organic farming with greater microbial biomass, increased species richness, enzyme activities and heterogeneity on a global scale [90,91]. Hartmann et al. found that organic farming increases richness, decreases evenness, reduces dispersion and shifts the soil microbial community structure when compared with conventionally farming [77]. In this study the quantity and quality of organic fertilizers are the main determinants of microbial biodiversity. Xia et al. revealed more diverse and abundant culturable endophytic fungal communities associated with organically managed crops such as corn, tomato, pepper, and watermelon [92]. All endophytic fungi isolated improved shoot growth and biomass of tomato plants, and some also exhibited activity that enhanced tomato fruit yields. Visioli et al. found that Tritordeum cv., a novel hexaploid cereal, is more efficient at hiring beneficial microbes with plant growth promoting capacity like Bacteroides under organic management [82]. Similarly, Li et al. found that under organic management, pepper plants recruit beneficial microbes more efficiently than under integrated or conventional management [78]. Specifically, the recruitment of *Bacillus* sp. reduces pepper blight (*Phytophthora capsici*) incidence and increases soil disease suppression under greenhouse conditions. Indeed, it is widely assumed that in organic farming systems. the presence of a great microbial biomass and high soil biodiversity can control the spread and increase of pest populations. Soil organisms can contribute to the control of soil-borne pathogens through nutrient competition, direct parasitism and direct inhibition by producing antibiotic metabolites and even by inducing plant disease resistance [93].

However, according to other authors there exists no difference, or less diversity between organic and conventional farming [80,90]. This may be due to the fact that organic farming often adopts intensive agricultural practices commonly utilized in conventional farming, which may hide the beneficial influence of lower agrochemicals use [94]. In addition, differences between these farming practices depend on land use type (arable, orchards, grassland), plant life cycle (annual and perennial) and climatic zone [90,91]. Measures to preserve and enhance biodiversity should therefore be more landscape- and farm-specific. By comparing microbiomes around different crops (wheat, barley, potato, carrot and lily) on the same soil, [79] showed that organic practices promote microbial diversity, richness and community heterogeneity compared to conventional agriculture. However, the scholars concluded that microbial communities respond to farming practices in varied and complex ways, and that increasing soil biodiversity does not necessarily entail improving soil health and plant productivity. In these cases, -Omics technologies such as metabolomics and metaproteomic, allow us to better detect the complexity of the soil microbial communities, the functional diversity and the metabolic pathways related to plant growth and health, hence, reducing this knowledge gap. Moreover, it is still unclear whether organic management practices *per se* have any direct benefit on the provision of ecosystem services. Nonetheless, the growing evidence for positive impact of organic farming demonstrates that this appears to be a promising strategy in the reduction of soil biodiversity loss and associated ecosystem services, especially food provision, with a simultaneous reduction in dependency on external inputs.

#### 3.1.2. Conservation Agriculture

It is a well-recognized fact that management characterized by no-tillage, crop rotation and/or intercropping, often with legumes, mineral fertilizers, and organic amendments (crop residues, compost and green manure), and maintenance of non-productive (‘natural’) elements are used to restore lost soil–plant system nutrients, improve soil structure, soil aggregation and water retention thanks to higher OM content. This leads to increased microbial species richness and overall density, microbial functional group diversity and complexity of the soil food web, with potential benefits for ecosystem functioning [94,95,96]. Enhancing crop diversity spatially (e.g., intercropping), or temporarily (e.g., crop rotation or cover crops), or both (e.g., agroforestry), has proven to increase the abundance of soil biota and species diversity, with beneficial effects on crop yield; this also reduces the development of pests and pathogens, decreasing the need to use chemical pesticides and fungicides [12,97,98]. Moreover, growing leguminous cover crops enhances microbial diversity due to an improved quality of residue input, meaning increased levels of N and C thorough N fixation and rhizodeposition respectively, thereby increasing soil fertility [99].

Minimum tillage or conservation agriculture and maintenance of crop residue cover on the soil surface benefit belowground food webs and processes, and improve abundance and diversity of soil bacterial communities, including beneficial microbes like *Pseudomonas, Burkholderiales* and *Rhizobiales* [83,100] with plant growth promoting capacity [101] and arbuscular mycorrhizal fungi, with beneficial effects on crop yield and biocontrol. This may be due to the fact that tillage-induced soil disturbances are eliminated, erosion losses are minimized, and large quantities of roots and above-ground biomass are returned to the soil, thereby conserving soil water and improving soil structure and organic matter content. This also increases agricultural adaptation to climate change and ecosystem service provisioning [102]. Conservation agricultural practices need to be adopted in combination with crop rotation or residue retention to both favor carbon sequestration and maintain or increase crop yields, particularly in dryland agricultural systems [103,104]. To sum up, agricultural practices that enhance maintenance and conservation of soil biodiversity also tend to promote the delivery of multiple ecosystem services.

### 3.2. Targeted Approach

Targeted manipulations of soil community composition are used to mitigate the negative environmental impact of agrochemicals and improve crop nutrient use efficiency, thereby reducing the need for some fertilizers, and leading to both enhanced crop yield and increased plant resilience to environmental stress [15].

The use of plant growth-promoting rhizobacteria (PGPR) and arbuscular mycorrhizal fungi (AMF) has been considered a promising alternative to conventional fertilizers to obtain high yields in a sustainable way, especially under stress conditions [105]. Rhizobacteria are microorganisms which naturally live in soils and form symbiotic interactions with plant roots, promoting plant health and productivity through: (a) plant hormone production such as auxins, cytokinin, and gibberellins, and inhibition of ethylene production; (b) symbiotic N₂ fixation; (c) inorganic phosphate solubilization and mineralization of organic phosphate and/ or other nutrients; (d) siderophores production, which enhances Fe bioavailability to plant; (e) antagonism towards phytopathogenic microorganisms through the production of siderophores, the synthesis of antibiotics, enzymes and/or fungicidal compounds, and the competition with detrimental microorganisms [106]. On the other hand, AMF are able to establish symbiotic interactions with the roots of 80% of plant families with consequent beneficial effects that can be explained in different ways: (a) improvement of soil structure through the binding action of the mycelial network and the release of glomalin, which contribute to soil stability and water retention, thereby increasing plant resistance to drought; (b) increase in nutrients (especially P, N, S, K, Ca, Fe, Cu and Zn) and water use efficiency, and (c) protection of plants against root pathogens [107]. PGP bacteria/rhizobacteria and AM propagules can be introduced by seed or soil inoculation [108], and soil or foliar spraying [109], as well as by using inoculum carriers like biochar, a porous structure that acts as a shelter for soil beneficial microorganisms by protecting them against grazers or competitors [110].

For example, the inoculation of maize plants with *Pseudomonas alcaligenes, Bacillus polymyxa*, and *Mycobacterium phlei* [111], and the inoculation of lettuces with *Trichoderma virens* or *Trichoderma harzianum* fungi [112] significantly promotes plant growth, nutrient uptake and nutrient content when grown under low soil nutrient levels. Application of PGPR and AMF consortia as well as of PGPR and N-fixing bacteria (*Azospirillum* spp., *Azoarcus* spp. and *Azorhizobium* spp.) improves plant growth and nitrogen accumulation in wheat, and increases plant resilience to environmental stresses due to the synergistic interactions between microorganisms [113,114]. In addition, they mitigate N losses in agricultural ecosystems, thereby reducing the pressure on the environment from the application of chemical fertilizers [113]. However, several biotic and abiotic factors may impact on the ability of plant-aiding microorganisms to successfully colonize the rhizosphere. Therefore, PGPM should tolerate nutrient- or water-limited conditions, be affine for root exudates, and compete with the resident rhizobacteria by secreting antibiotics [76]. For instance, [74,115] investigated the action of plant-derived protein hydrolysates (PHs) biostimulant and the effects of the mycorrhizal inoculation on morpho-physiological traits and metabolic profile of tomato and durum wheat respectively, grown under limited water availability. PHs and AMF biostimulant had positive effects on biomass, phytohormones production and improved tolerance to reactive oxygen species (ROS)-mediated oxidative imbalance. Likewise, [75] demonstrated a clear link between endophytic fungi - mediated yield increase in pepper plants and a strong up-regulation of phytohormones. Moreover, metabolomics analysis highlighted the molecular basis of the improved loquat plant growth in the presence of the root rot fungus *Armillaria mellea* following AMF colonization [116]. Recent studies have investigated the effects on the rhizosphere microbial communities of different types of microbial consortia (consisting of PGPRs alone or in association with AMFs) and biostimulant substances coated on crop seeds. All microbial consortia and biostimulant substances significantly enhance plant growth and nitrogen accumulation in wheat and maize, without affecting the endogenous microbial diversity or the composition of the rhizospheric microbial community, rather stimulating rare bacterial taxa, some of which involved in plant growth and pathogen resistance [61,76,81]. This suggests that seed-applied biofertilizers may be exploited effectively in sustainable cultivation. However, since the inoculation with PGP bacteria and AM fungi as biofertilizers entails important issues, such as short shelf-life, on-field viability, variable performance under fluctuating environmental conditions (temperature, radiation, pH sensitive), and especially the need for large fields, further research is needed. At this time, despite the extensive research over the last ten years on the use of PGPM application, only a few studies consider in situ PGPM monitoring after their application, although their abundance, activity and inoculation schedule are important parameters to be considered to maximize crop yields [117,118]. Culture-dependent approaches are still commonly utilised to estimate the persistence of the bioinoculants in the rhizosphere or endosphere environment, but the major limitation of this analysis is the difficult to differentiate the inoculated organisms from native populations based on morphological characteristics. To date, PGPM tracking methods have included the use of microscopy-based techniques and the use of reporter genes, such as *gus A*, *lux* and GFP, for single cell detection and semi-quantitative in situ detection (especially for endophytes), immune-associated semi quantitative techniques as enzyme-linked immunosorbent assay (ELISA), fluorescence microscopy and nucleic acid-based systems, such as fluorescence in situ hybridization (FISH), community analysis techniques (DGGE, T-RLFP, ARISA, ARDRA) and qPCR. For a comprehensive review on these methodologies, see [118,119]. In this regard, the development of protocols for amplification and quantification of nucleic acids even when present at very low number, a droplet digital PCR application could be a valid solution for the detection of inoculated PGPM to soil samples [117]. In addition, -Omics analyses, in particular NGS analyses, although much more expensive, are also very useful and rapid methodologies for a relative quantitative evaluation of the microorganism or microorganism consortia applied as biofertilizers on root tissues and in the rhizosphere soil, starting from the first stage of plant development to the end of vegetative growth [61,76,81]. Moreover, the determination of the entire genomic DNA sequence (WGS) of a microbial strain could also be a powerful approach to investigate the survival efficiency in the soil/rhizosphere of the inoculum [119]. WGS could also be used in combination with shot gun sequencing to identify microbial strains in the soil metagenome and to evaluate natural occurrence of beneficial microorganisms in different soil and rhizosphere environments [119,120].

## 4. Implications of Soil Biodiversity for Nutrition and Food Safety

“Soil is where food begins” (Food and Agriculture Organization (FAO), Rome, Italy). This means that through the wide range of ecosystem services that it provides, soil is an essential component in sustaining food security. The Food and Agriculture Organization (FAO) stated that ‘‘food security exists when all people, at all times, have physical, social and economic access to sufficient, safe and nutritious food, which meets their dietary needs and food preferences for an active and healthy life”. Food security has four dimensions: (1) food production and availability through agronomic management of soil resources; (2) stability of food production and availability at all times; (3) food access through economic and physical capacity of households or communities; and (4) food safety and utilization through nutritious and biological quality.

### 4.1. Food Nutritional Properties

Human nutrition ultimately depends on the availability and balance of different nutrients in soil and on the ability of plants to extract those nutrients. Agricultural products must provide about 50 nutrients essential to human health (e.g., vitamins, minerals, trace elements, amino acids, essential fatty acids), 7 macrominerals (Na, K, Ca, Mg, S, P, Cl), and 15 microelements (Fe, Zn, Cu, Mn, I, F, B, Se, Mo, Ni, Si, Li, Sn, V, Co), which must be supplied through soil [121]. However, certain nutrient limitations in soil, or the absence of belowground interactions that promote nutrient uptake, can result in ‘hidden hunger’, i.e., specific nutritional deficiencies in the food produced. Over two billion people in the world suffer from micronutrient deficiencies, which give rise to dangerous health conditions and diseases, such as birth defect, cancer, cardiovascular conditions, osteoporosis, neurodegenerative disorders and mental health problems [122,123,124]. A solution to redress nutritional imbalances has been post-harvest fortification of food products, e.g., by adding essential B-vitamin, zinc or iron to flours [125]. Other important approaches that supplement staple food with micronutrients are through the use of chemical fertilizer and nano-fertilizer spraying, and agronomic biofortification, such as conventional breeding and genetic modification [124]. Additionally, microbe-mediated biofortification is a new and promising alternative to increase bioavailability of plant nutrients [124]. Biofortification through microbes is obtained by applying various targeted and untargeted management methods to increase soil microbial diversity that solubilize the essential soil minerals and micronutrients and are made easily available for uptake by the plants. Thus, various plant growth-promoting microbes (PGPM) including bacteria, fungi, Cyanobacteria, Actinobacteria and mycorrhiza are used to improve plant performance and nutrient content [124] (Table 2). 

In particular, endophytic microbes appear to be more efficient than rhizospheric microbes due to the fact that they are present in the plant and interact with it more closely compared to rhizosphere, resulting in better health, greater crop production, and enhanced nutritional value [124]. Moreover, since AMF form the most widespread plant-microbe symbiosis and interact with almost all edible crops, they are the most studied in regard to nutrient level in edible plant tissues [107]. 

Besides macro- and micro-nutrients, PGPM enhance plant provision of secondary metabolites (PSC) known as phytonutrients or nutraceuticals, which are essential to promote human health through reduction of oxidative damage, modulation of detoxifying enzymes, stimulation of the immune system, and prevention of chronic diseases, arteriosclerosis, heart diseases and cancer [128,135,136]. These phytonutrients include terpenoids such as carotenoids, alkaloids, polyphenols (phenolic acids, anthocyanins, flavonoids), glucosinolates, lignans, stilbenes, sulphur-containing compounds and vitamins [136,137]. In general, plants produce natural defense substances like PSC after biotic or abiotic stress exposure, mainly depending on plant genotype, but their expression may be modulated by various different agronomic and environmental factors, including crop interaction with rhizosphere microbes [135,136]. For instance, the metabolomic analyses disclosed a clear link between AMF and PGPR inoculation in pepper and basil respectively, increased crops yield and the accumulation of several PSC associated with crop defense to environmental stresses [75,138]. Similarly, the metabolome analysis revealed that olive growing in fields where no-tillage and crop residue cover were applied, shown higher PSC concentration in xylem sap [139]. Moreover, the proportion of essential amino acids and the contents of K, Mg, Na, Mn, Zn and tocopherol were higher in butternut squashes (*Cucurbita moschata*) growing in organic farming compared to conventional farming [140]. The presence of PGPM also increases the concentration of organic acids in fruits (like citric and malic acids), which play a critical role in maintaining the quality of a variety of food and contributing to its sensory properties like flavor. Fruit flavor and fragrance are determined by many compounds, such as sugars, acids and more than two hundred volatile constituents [127]. Untargeted metabolomics has been utilized even for evaluating coffee quality associated with polyphenols, alkaloids, diazines and Maillard reaction products and the potential correlations with the sensory attributes [141]. 

Other authors have reported improvements in food quality, which can be attributed to microorganisms. For instance, recent field research on the application of different soil microbial consortia to common wheat has highlighted heightened plant growth and nitrogen accumulation in the phases of stem elongation and heading, as well as upregulation of two gluten protein subunits, which are important indicators of flour quality [76,142]. Similarly, [82] demonstrated that the resilient cereal Tritordeum was better adapted to organic farming compared to conventional farming through an increased abundance of beneficial bacteria (Bacteroidetes) in the soil microbial community of the rhizosphere. The amount of minerals, gluten proteins and polyphenols increased in the grains too under this management, which would imply a link between belowground biodiversity and plant quality traits. The results of these studies suggest that an effective exploitation of beneficial microorganisms in sustainable agricultural practices, may be possible to enhance plant physiological status, increase yield quantity/ quality and ameliorate human health.

However, many nutraceutical properties of fresh fruits and vegetables are jeopardized [143]. In such cases, post-harvest measurements lose relevance because biodiversity-related effects may diminish before the consumer is reached. A unique study reported very specific results on the taste of some bread made from AM (complex inoculant) and non-mycorrhizal wheat [144]. This result is extremely encouraging considering the relative complexity of the operations (milling, mixing, yeast addition, baking) involved, but a greater number of similar studies are necessary to reach broader generalizations.

### 4.2. Food Safety

Around the world, over 420,000 people die and some 600 million people, almost one in ten, fall ill after eating contaminated food every year. Indeed, foodborne hazards are known to cause over 200 acute and chronic diseases, ranging from digestive tract infections to cancer [145]. 

In regards to food safety, research has focused on the role of microbes in reducing food contamination from heavy metals, agrochemicals, industrial and urban waste, as well as soil-borne pathogens. Beneficial microorganisms like PGPB/R and AMF are now used in agriculture to replenish the use of agrochemicals, minimize the negative impact on the environment, and increase the quantity, quality and safety of farm products with subsequent benefits on human health [15,93,105]. Contaminants can reach the soil by atmospheric deposition, waste disposal, industrial effluents and direct application, and afterwards, groundwater, streams and sea through soil washing [125,146]. Irrigation water and flooding events are also related to soil contamination [123]. Polluted soils affect crop yields and food safety, especially when contaminants bioconcentrate into organisms within food chains [123]. Contaminants reach the human body systems through three main pathways: (1) oral ingestion, (ii) breathing, and (iii) infiltration through the skin [137]. Pesticides persist in food, including vegetables, fruits, meat and in the human organism, causing severe illnesses ranging from respiratory disorders and musculoskeletal illnesses to dermal and cardiac related diseases which are more serious in farm handlers [137]. Trace metals like As, Pb, Cr, Hg and Cd, which are often contained in pesticides [146], accumulate in the topsoil due to their strong affinity with organic matter, and can passively be taken up by plants through water flow [123]. Pb and Cd cause several diseases, especially cardiovascular, kidney, nervous system, blood as well as bone diseases, and are considered potential carcinogens. Despite Zn and Cu being essential elements, food containing a high concentration of these metals is considered toxic to humans and animals [147]. 

Currently, many studies have shown a specific way by which microorganisms act as control agent to reduce contaminant concentration in food. For instance, the presence of rhizosphere microbes, especially AMF, can reduce heavy metal concentration in crops [125]. The ratios of inorganic/organic As concentrations in rice grains are significantly reduced by AMF, which involve the transformation of inorganic As into a less toxic organic form of dimethylarsenic acid (DMA) in rice [148]. Severe soil biodiversity loss has shown to increase the uptake of two insecticides (acetamipril and imidacloprid) by *Brassica chinensis* [149]. Furthermore, the AMF *Glomus irregulare* limited the growth of pathogen *Fusarium sambucinum* by inhibiting the production of mycotoxin trichothecene 4,15-diacetoxyscirpenol in roots and tubers of potato plants, which can lead to health problems in humans and animals [150]. Moreover, symbiotic plant microbes such as arbuscular mycorrhizal fungi have been shown to reduce storage-induced pests on staple crops such as potatoes [143]. 

Pathogen contamination of fresh produce is a leading worldwide threat to human health, underlying millions of illnesses, and thousands of deaths [151]. *Salmonella, Shigella* spp., *E. coli* and *Campylobacter* are the major pathogens that reach the human body through contaminated fresh vegetables usually grown in soils amended with non-composted animal manure [146]. Some of these pathogens, such as *Salmonella enterica* and shiga-toxin producing *Escherichia coli* O157:H7, survive longer in soils amended with cow slurry due to its higher soluble carbon and nitrogen contents compared with solid manure [152]. It is a common practice to remove hedgerows, ponds, or other natural habitats to reduce feces contamination by vertebrate wildlife in fruit or vegetables, and therefore preserve food safety. However, recent evidence had suggested that land simplification increases the probability of food contamination with human pathogens. The authors of [151] found that organic farms can shelter a diverse community of feces-feeding beetles and microbes that suppress the human pathogen *Escherichia coli* more efficiently than conventional farming by removing animal feces once deposited. Furthermore, [153] found that *Salmonella enterica* Typhimurium introduced into tomato plants grown in organically or conventionally managed soil lived longer in leaves with limited microbial communities than in leaves with a diverse endophytic microbiome. 

Little is currently known about the role of soil microbial diversity on the control soil-transmitted human pathogens. However, it is assumed that the more diverse and complex the soil microbial community is, the higher the competition for nutrients, which inhibits the development or persistence of pathogens in the soil. Lastly, soil microbial diversity brings direct benefits for human health as it can suppress disease-causing soil organisms, provide clean air, water and food, and is a source for producing antibiotics [125].

## 5. How Beneficial Soil Microbes, Food and Gut Are Interconnected

Microbiomes found in soil, plants and humans are interconnected: gut and soil microbiome have similar bacteria phyla (Firmicutes, Bacteroidetes, Proteobacteria, Actinobacteria), and fruits, salads and vegetables contain microbes which add to the human gut microbiome. Thus, the soil-plant microbiome can have an effect on the gut microbiome and thereby human health [16,154]. Moreover, root and gut microbes seem to have similar functions since they are involved in nutrient absorption, host gene expression, disease resistance and seem to share evolutionary trends [155]. Root microbes are able to synthesize phytohormones, siderophores and antibiotics, while plant and gut microbes synthesize fundamental amino acids, vitamins and numerous other secondary metabolites that act on their host immune system. Indeed, the soil-plant and gut microbiomes are to be seen as meta-organs of utmost importance for their hosts’ health [16].

Studies on several mammalian herbivores, omnivores and carnivores have shown that the gut microbial community is very dynamic and comprises both autochthonous and allochthonous members absorbed through food and water, and also by direct contact with the environment/soil in which they live [154,156,157]. For instance, gfp-tagged enteric pathogens were proven to transfer from cattle feed into the intestinal tract of cows and their manure, then into soil, cress plants, and snails and their excrements [152]. High bacterial diversity is exhibited in herbivore guts, including plant-associated species such as endophytes. Endophytes reside in plant tissues, therefore may survive stomach digestion [156]. Moreover, human gut communities appear to compare with other omnivores, and are most closely related to the Bonobos or Pygmy chimpanzees, which feed on fruits [158]. The intestinal microbiome has changed during human evolution, thereby reflecting the modifications in food supply and lifestyle: modern lifestyle is characterized by improved hygiene, consumption of processed food and the extensive use of drugs, notably antibiotics, and this appears to have impacted on human gut microbiome diversity in the past decades, overall lessening its variety [159]. These changes are still visible between urban and rural communities [160]. The absence of contact with outdoor-associated natural beneficial microbiota has an indirect effect on the human gut microbiome, with possible negative consequences on human health [159]. Recent studies have supported the “hygiene hypothesis”, which suggests that environments displaying rich microbial diversity provide protection against allergies and autoimmune disorders [161]. Indeed, besides the many different functions which the intestinal microbiome serves in human health, it is clear that it also plays a part in numerous gastrointestinal (GI) and non-gastrointestinal diseases, such as obesity/metabolic syndrome, atherosclerosis/cardiovascular diseases and neurologic/psychiatric diseases [155]. For example, recent research using mice proved that gut microbes acquired from soil heightened anti-inflammatory capacity to TH2-type inflammation responses in comparison to mice who had had no soil contact [162]. Other studies using mouse models have also shown that exposure to *Mycobacterium vaccae*, a common soil saprophyte, is involved in both immune system activation and specific serotonergic pathways that act on emotional and behavioral response to stress [163,164].

Therefore, a healthy diet, i.e., one that contains plenty of fiber, minerals and vitamins from fruits and vegetables, and a closer contact with the natural environment may preserve gut microbiome richness [154]. However, in order to safeguard human gut microbiome diversity to favor a healthy life, we should also consider the soil management practices applied in the fields which our food comes from. Indeed, the current food production system is mainly based on extensive monocultures of a few selected crop varieties that require fertilizers, herbicides and pesticides to ensure a high yield, resulting in poor microbial diversity in the soil [94]. Soil erosion and climate change also reduce microbial diversity and lead to the loss of vast areas of arable land and their microbial populations [16]. Therefore, the reduced presence of important symbiotic partners of main food crops have lowered the production of vitamins, minerals, antioxidants and other metabolites that are beneficial to both plants and human health. For example, the use of plant breeding to reduce the bitterness of Brassicaceae, such as broccoli, cauliflower and cabbage, has led to a decrease in the production of glucosinolates [156]. This secondary metabolite helps the plant resist pathogens and is considered an anti-cancer metabolite in humans [156]. Moreover, the extensive use of the glyphosate herbicide has shown negative effects on beneficial soil, rhizosphere and endospheric microbes, including AMF and nitrogen-fixing *Rhizobium* spp., and even on humans, since it is considered a potential carcinogen with possible negative effects on the gut microbiome [165]. Several other pesticides, such as carbamates, pyrethroids and neonicotinides, have also shown negative effects on beneficial microbes, with direct and indirect implications on soil, plant and food safety, and human health [95]. 

Many fresh fruits, salads and vegetables are treated with various pesticides and antibiotics to preserve them during the stored and shipping period. Some of these chemicals reduce the presence of beneficial plant microbes and through food ingestion also affect the gut microbiomes. Furthermore, the application of manure from antibiotic-treated animals in cultivated fields impacts on the microbial functions and composition of soil, so the consumption of fresh produce from these lands can extend resistance genes to the human gut microbiome and favor the emergence of multi-drug-resistant human pathogens [16]. Similarly, the extensive use of pesticides and herbicides could increase the risk of new pathogens and diseases against both plants and humans [16]. The use of beneficial microorganisms in agriculture contributes to providing healthy food by limiting the use of fertilizers and pesticides; at the same time, by eating organic food and unprocessed organic products, humans could benefit from both the intake of nutritionally important secondary metabolites and the intake of microorganisms that can be useful for the human intestine [12]. Research on the role of the microbiome on human metabolism and health should therefore not be limited to the human gut microbiome but should include plant microbiota and its role in the growth and development of edible plants as such an effort should for the benefit of both plants and humans. For instance, [166] recommends consuming fresh land-based (nature-based) food for adequate human microbiome, which derives from nature based-agriculture. 

Future research on soil microbial communities and human health should focus on integrating soil ecology and agronomic crop production with human health and nutritional sciences. It is therefore essential to improve our understanding of the functions and roles of the hundreds of different microbial species in the complex interaction network with their hosts, and to identify the best farming practices that through microbial diversity manipulation provide safe and nutritious food with high yield levels. The different -Omics can now be used to reach this goal; these could be of great help in further understanding and restoring human health and the functioning of ecosystems, which are currently under widespread pressure. New high throughput –OMICS analyses could clarify the interconnection, proposed by the One Health concept, between the health conditions of organisms, communities and ecosystems through microbial communities at different levels of integration in time and space [154], with the aim of implementing programmes, policies, legislation and research.

## Figures and Tables

**Figure 1 microorganisms-09-01400-f001:**
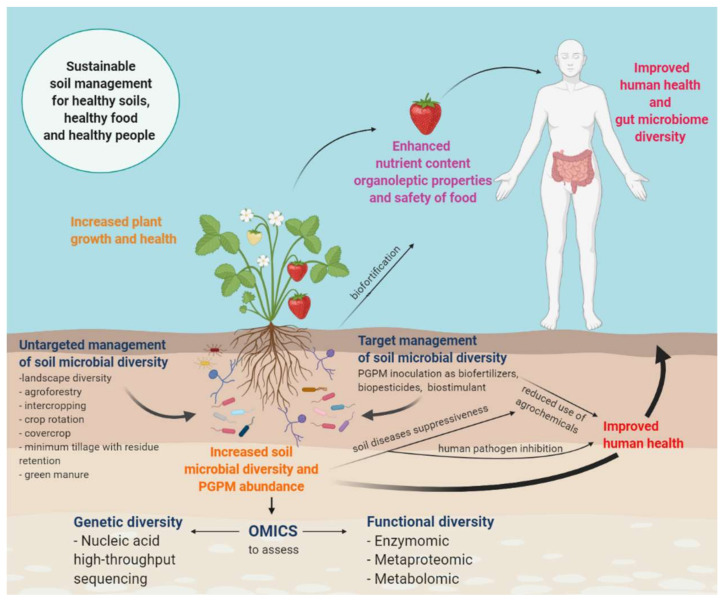
Both targeted and untargeted management of soil microbial diversity in the sustainable improvement of plant health, food crop yield, its nutritional quality and safety. The development of high-throughput technologies, called -Omics, applied to the study of soil microbial functional diversity will help strengthen the link between soil well-being, food quality, food safety and human health.

**Table 2 microorganisms-09-01400-t002:** Examples of studies showing the effect of rhizosphere microbes on food nutritional quality.

Crop	Rhizosphere Microbes	Plant Tissue	Impact on Food Quality and Nutritional Status	Reference
Lettuce	*Azotobacter chroococcum* and *Glomus fasciculatum; G. fasciculatum* and *Glomus mosseae*	Leaves	Increased concentration of total phenolic compounds, anthocyanins, and carotenoids; increased flavonoid content	[126]
Tomato	*Pseudomonas* strains and a mixed mycorrhizal inoculum, under conditions of reduced fertilization	Fruit	Larger and better-quality fruits (concentration of sugars, organic acids, such as citric and malic acid, and vitamin C)	[127]
Strawberry	*Bacillus, Pseudomonas*	Fruit	Increased fruit yield, plant growth, nutrient content (P, Fe, Zn, and vitamin C)	[106]
Tomato	AMF *G. intraradices*	Fruit	Increased plant growth, mineral nutrient content (P, K, Ca, Zn), and enhanced nutritional and nutraceutical value (carotenoids such as lycopene)	[128]
Spinach	AMF and bacterial species	Leaves	Higher concentration of total phenolic compounds, flavonoids, and phenolic acid	[129]
Highbush blueberry	PGPR (*Pseudomonas* sp. and *Bacillus* sp.) and AMF (*Gliocladium virens* and *Trichoderma harzianum)*	Leaves	Increased plant growth and enhanced P, Zn and Cu uptake	[130]
Strawberry	AMF and selected *Pseudomonas* strains, under conditions of reduced fertilization	Fruit	Increased concentration of antioxidant molecules (anthocyanins). Yield not affected	[131]
Wheat grains	PGPB (*Providencia* sp.) and cyanobacterial strains (*Anabaena* sp., *Calothrix* sp. and *Anabaena* sp.)	Grain	Increased yield, and micronutrient (Fe, Zn, Cu, Mn) and protein enrichment	[132]
Rice grains	Endophytic strains, *Burkholderia* sp., *Sphingomonas* sp., *Variovorax* sp., and *Enterobacter* sp.,	Grain	Improved plant growth, yield and root morphology, increased Zn concentration in shoot and roots	[133]
Wheat grains	Endophytic bacteria, *Enterobacter* sp. and *Burkholderia phytofirmans*	Grain	Enhanced Fe concentration, plant height, leaf area, spike length, and plant biomass	[134]

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
