# Peer review of "Improvement of Soil Microbial Diversity through Sustainable Agricultural Practices and Its Evaluation by -Omics Approaches: A Perspective for the Environment, Food Quality and Human Safety"

_microorganisms, 2021, doi:10.3390/microorganisms9071400_

Round 1
Reviewer 1 Report
Comment #1: Lines 28-29: “Twenty-first century global issues like food security, demand for energy and water, climate change and biodiversity are in part associated with the sustainable use of soils.”
It does not seem reasonable. Sustainable use of soil is positive and should not cause the above global issues.
Comment #2: Lines 43-52
This sentence is incredibly long and hard for readers to follow. Please rephrase it and make it more readable.
Comment #3: Lines 52-55 “Nevertheless, in recent years several studies have shown that anthropogenic activities, such as agricultural intensification and land use change, as well as climate change, are adding stress on the ability of soil to sustain its important role in the planet’s survival [9].”
What do you mean by the planet’s survival? The words “adding stress on the ability of soil to sustain its important role” can be replaced by better expression. You mentioned several studies but only have one reference here. Please add more supporting references here.
Comment #4: Lines 70-74 “In addition, some processes that provide ecosystem functions are carried out by a greater number of species or groups of organisms (e.g. organic matter degradation), whereas other processes tend to be carried out by fewer species or groups of organisms (e.g. atmospheric nitrogen fixation) and so 73 are more easily compromised by ecological disturbances [18].”
This sentence is too long. Please rephrase it.
Comment #5: Lines 77-81 “In this regard microbial communities offer a great potential to assess the functional biodiversity in soils because they are ubiquitous, are actively involved in biological functioning and ecosystem services provisioning, and are highly sensitive to environmental changes in terms of modifications in biomass, structure/diversity and activity [20].”
More references are needed here. Please consider the following:
Roy, K., Ghosh, D., DeBruyn, J.M., Dasgupta, T., Wommack, K.E., Liang, X., Wagner, R.E. and Radosevich, M., 2020. Temporal dynamics of soil virus and bacterial populations in agricultural and early plant successional soils. Frontiers in Microbiology, 11, 14194.
Liang, X., Zhuang, J., Löffler, F.E., Zhang, Y., DeBruyn, J.M., Wilhelm, S.W., Schaeffer, S.M. and Radosevich, M., 2019. Viral and bacterial community responses to stimulated Fe (III)‐bioreduction during simulated subsurface bioremediation. Environmental microbiology, 21(6), 2043-2055.
Comment #6: Line 104 “the advent of next generation sequencing techniques as complete genome”
“the advent of next generation sequencing techniques such as complete genome”
Comment #7: Lines 107-108 “while the other –Omics approaches have allowed the investigation of these communities’ biological functions [22].”
What are included when you say the other –omics approaches?
Comment #8: Lines 108-110 “Soil microbial ecology nowadays is not just describing with high resolution the soil microbial community structure but embraces approaches and tools able to link structure and function [23].”
The sentence seems not right. Please rephrase.
Comment #9: Lines 110-112 “Some important ecological functions are linked to biogeochemical cycles (e.g. nitrogen fixation, litter decomposition, plant productivity) and to their interactions with the soil food web [24].”
What do you mean by ecological functions? Don’t they include biogeochemical cycles and soil food web?
Comment #10: Line 116 “in different field application”
“different fields of applciation”?
Comment #11: Lines 114-117 “Different authors advocated for next-generation experiment linking community structure to ecological processes in different field application (e.g. agriculture, bioremediation, industry) [26,27] and ultimately to define the importance of SBD in ecosystem process, function and services [28].”
Not clear. Needs to be rephrased.
Comment #12: Lines 122-123 “The analysis of 16S-rDNA and ITS sequences to assess the bacterial and fungal community composition of a soil sample is a technique whose wide use has increased with the advent of high-throughput sequencing.”
Very redundant.
Comment #13: Line 127 “bacterial 16S rDNA gene”
Sholud be “bacterial 16S rRNA genes”
Comment #14: Lines 135-137 “an emerging molecular method which enables linking the community structure with possible soil functional processes, thereby allowing for the study of functional microbial diversity [30].”
Grammar errors here. Please rephrase.
Comment #15: Lines 247-249 “It has been suggested that the integrated management of microbial SBD can be a promising strategy to obtain agricultural systems which are more productive, resource-efficient, resilient and adaptive to global changes, while minimizing environmental impacts [11].”
The structure of the sentence is strange which is not quite the style for scientific publishing. Please rephrase it.
Comment #16: Lines 250-255 “Several studies have highlighted that is possible and advantageous to address future needs by transitioning from conventional intensification of agriculture to a food production system based on “ecological intensification”; this means that microbial SBD can be effectively exploited as a nature-based solution to maintain high productivity levels by optimizing soil ecosystem services, while reducing the reliance on external input and minimizing adverse effects on the environment [67–70].”
How can soil biodiversity be a nature-based solution? Soil biodiversity can be an index but not a solution.
Comment #17: Lines 292-293 “Data on soil microbial biodiversity in organic and conventional farming are rather controversial.”
How can data be controversial?
Comment #18: Line 578 “5. How beneficial microbes in the soil, food and gut are interconnected”
Suggestion “5. How beneficial soil microbes, food and gut are interconnected”
Author Response
Reviewer #1
Comment #1: Lines 28-29: “Twenty-first century global issues like food security, demand for energy and water, climate change and biodiversity are in part associated with the sustainable use of soils.”
It does not seem reasonable. Sustainable use of soil is positive and should not cause the above global issues.
Answer to Reviewer: the sentence was eliminated to avoid misunderstanding
Comment #2: Lines 43-52
This sentence is incredibly long and hard for readers to follow. Please rephrase it and make it more readable.
Answer to Reviewer: the sentence was eliminated do avoid redundancy
Comment #3: Lines 52-55 “Nevertheless, in recent years several studies have shown that anthropogenic activities, such as agricultural intensification and land use change, as well as climate change, are adding stress on the ability of soil to sustain its important role in the planet’s survival [9].”
What do you mean by the planet’s survival? The words “adding stress on the ability of soil to sustain its important role” can be replaced by better expression. You mentioned several studies but only have one reference here. Please add more supporting references here.
Answer to Reviewer: the sentence was rephrased and a reference was added (European Environment Agency. Land and Soil in Europe: why we need to use this vital and finite resurces sustainably? Pubblcation. Office Luxemburg 2019)
Comment #4: Lines 70-74 “In addition, some processes that provide ecosystem functions are carried out by a greater number of species or groups of organisms (e.g. organic matter degradation), whereas other processes tend to be carried out by fewer species or groups of organisms (e.g. atmospheric nitrogen fixation) and so 73 are more easily compromised by ecological disturbances [18].”
This sentence is too long. Please rephrase it.
Answer to Reviewer: the sentence was rephrased
Comment #5: Lines 77-81 “In this regard microbial communities offer a great potential to assess the functional biodiversity in soils because they are ubiquitous, are actively involved in biological functioning and ecosystem services provisioning, and are highly sensitive to environmental changes in terms of modifications in biomass, structure/diversity and activity [20].”
More references are needed here. Please consider the following:
Roy, K., Ghosh, D., DeBruyn, J.M., Dasgupta, T., Wommack, K.E., Liang, X., Wagner, R.E. and Radosevich, M., 2020. Temporal dynamics of soil virus and bacterial populations in agricultural and early plant successional soils. Frontiers in Microbiology, 11, 14194.
Liang, X., Zhuang, J., Löffler, F.E., Zhang, Y., DeBruyn, J.M., Wilhelm, S.W., Schaeffer, S.M. and Radosevich, M., 2019. Viral and bacterial community responses to stimulated Fe (III)‐bioreduction during simulated subsurface bioremediation. Environmental microbiology, 21(6), 2043-2055.
Answer to Reviewer: We inserted the references according to the suggestion
Comment #6: Line 104 “the advent of next generation sequencing techniques as complete genome”
“the advent of next generation sequencing techniques such as complete genome”
Answer to Reviewer: DONE
Comment #7: Lines 107-108 “while the other –Omics approaches have allowed the investigation of these communities’ biological functions [22].”
What are included when you say the other –omics approaches?
Answer to Reviewer: we intended the other approaches mentioned (transcriptomics, proteomics, enzymomics and metabolomics). Now this was stated in the sentence
Comment #8: Lines 108-110 “Soil microbial ecology nowadays is not just describing with high resolution the soil microbial community structure but embraces approaches and tools able to link structure and function [23].”
The sentence seems not right. Please rephrase.
Answer to Reviewer: the sentence was removed
Comment #9: Lines 110-112 “Some important ecological functions are linked to biogeochemical cycles (e.g. nitrogen fixation, litter decomposition, plant productivity) and to their interactions with the soil food web [24].”
What do you mean by ecological functions? Don’t they include biogeochemical cycles and soil food web?
Answer to Reviewer: the sentence was removed to avoid redundancy
Comment #10: Line 116 “in different field application”
“different fields of applciation”?
Answer to Reviewer: the phrase was modified
Comment #11: Lines 114-117 “Different authors advocated for next-generation experiment linking community structure to ecological processes in different field application (e.g. agriculture, bioremediation, industry) [26,27] and ultimately to define the importance of SBD in ecosystem process, function and services [28].”
Not clear. Needs to be rephrased.
Answer to Reviewer: we modified the sentence
Comment #12: Lines 122-123 “The analysis of 16S-rDNA and ITS sequences to assess the bacterial and fungal community composition of a soil sample is a technique whose wide use has increased with the advent of high-throughput sequencing.”
Very redundant.
Answer to Reviewer: we eliminate the first sentence to avoid redundancy
Comment #13: Line 127 “bacterial 16S rDNA gene”
Sholud be “bacterial 16S rRNA genes”
Answer to Reviewer: DONE
Comment #14: Lines 135-137 “an emerging molecular method which enables linking the community structure with possible soil functional processes, thereby allowing for the study of functional microbial diversity [30].”
Grammar errors here. Please rephrase.
Answer to Reviewer: DONE
Comment #15: Lines 247-249 “It has been suggested that the integrated management of microbial SBD can be a promising strategy to obtain agricultural systems which are more productive, resource-efficient, resilient and adaptive to global changes, while minimizing environmental impacts [11].”
The structure of the sentence is strange which is not quite the style for scientific publishing. Please rephrase it.
Answer to Reviewer: DONE
Comment #16: Lines 250-255 “Several studies have highlighted that is possible and advantageous to address future needs by transitioning from conventional intensification of agriculture to a food production system based on “ecological intensification”; this means that microbial SBD can be effectively exploited as a nature-based solution to maintain high productivity levels by optimizing soil ecosystem services, while reducing the reliance on external input and minimizing adverse effects on the environment [67–70].”
How can soil biodiversity be a nature-based solution? Soil biodiversity can be an index but not a solution.
Answer to Reviewer: you are wright; we change “SBD” with “soil microorganism enrichment”
Comment #17: Lines 292-293 “Data on soil microbial biodiversity in organic and conventional farming are rather controversial.”
How can data be controversial?
Answer to Reviewer: We rephrased the sentence to explain better the concept
Comment #18: Line 578 “5. How beneficial microbes in the soil, food and gut are interconnected”
Suggestion “5. How beneficial soil microbes, food and gut are interconnected”
Answer to Reviewer: We modified according to the suggestion
Reviewer 2 Report
This is an extensive and interesting review of the role of soil microbial diversity mainly in agricultural ecosystems. Despite the title, the work not only presents the omics techniques most used in the study of SBD, but also summarizes the main results obtained in various studies on the role of SBD in agricultural production, the nutritional quality of products and human health. Perhaps the title could better reflect this content focusing less on omic-approach and more on the management of SBD
There is only one block of information that I have missed. In the section "3.2 Targeted approach" where it talks about biofertilizers, biostimulants, etc, the difficulty of monitoring microorganisms once applied to crops is not mentioned. Currently, one of the main challenges of using these products is to be able to relocate them later either on the soil or in the plant. In this sense, I recommend reviewing works such as Kokkoris, V., Vukicevich, E., Richards, A., Thomsen, C., & Hart, M. M. (2021). Challenges Using Droplet Digital PCR for Environmental Samples. Applied Microbiology, 1 (1), 74-88. I suggest including a short paragraph to reflect the difficulties to detect the microorganisms once they were artificially applied in agroecosystems.
Author Response
Reviewer #2
This is an extensive and interesting review of the role of soil microbial diversity mainly in agricultural ecosystems. Despite the title, the work not only presents the omics techniques most used in the study of SBD, but also summarizes the main results obtained in various studies on the role of SBD in agricultural production, the nutritional quality of products and human health. Perhaps the title could better reflect this content focusing less on omic-approach and more on the management of SBD
There is only one block of information that I have missed. In the section "3.2 Targeted approach" where it talks about biofertilizers, biostimulants, etc, the difficulty of monitoring microorganisms once applied to crops is not mentioned. Currently, one of the main challenges of using these products is to be able to relocate them later either on the soil or in the plant. In this sense, I recommend reviewing works such as Kokkoris, V., Vukicevich, E., Richards, A., Thomsen, C., & Hart, M. M. (2021). Challenges Using Droplet Digital PCR for Environmental Samples. Applied Microbiology, 1 (1), 74-88. I suggest including a short paragraph to reflect the difficulties to detect the microorganisms once they were artificially applied in agroecosystems.
Answer to Reviewer: We thank the reviewer for the interest in the paper and for the positive evaluation. According to his/her suggestions we modified the title of the review as follows:
“Improvement of soil microbial biodiversity through sustainable agricultural practices and its evaluation by –Omics approaches: a perspective for the environment, food quality and human safety”
In addition we inserted in the section 3.2 “Targeted approach” a final short paragraph concerning the difficulties to detect the microorganisms after their application in agrosystems and the methods that can be applied at this purpose.
Round 2
Reviewer 1 Report
My concerns have been addressed.
Author Response
We thank the reviewer for the positive evaluation of the manuscript